# PAXIP1-AS1 is associated with immune infiltration and predicts poor prognosis in ovarian cancer

**Buze Chen** [1,2☯]*, **Xiaoyuan Lu** [1,2☯], **Qingmei Zhou** [1], **Qing Chen** [1], **Siyan Zhu** [1], **Guilin Li** [3], **Hui Liu** [4]*

**1** Department of Gynecology, The Affiliated Hospital of Xuzhou Medical University, Xuzhou, China, **2** Xuzhou Medical University, Xuzhou, China, **3** Department of Gynecology, Maternal and Child Health Care Hospital Affiliated to Xuzhou Medical University, Xuzhou, China, **4** Department of Pathology, The Affiliated Hospital of Xuzhou Medical University, Xuzhou, China

☯ These authors contributed equally to this work.
* hliu@xzhmu.edu.cn (HL); zku4ba@163.com (BC)

**Data Availability Statement:** The data used in this study were from TCGA-OC (https://portal.gdc.cancer.gov/projects/TCGA-OV) and GSE138866 (https://www.ncbi.nlm.nih.gov/geo/query/acc.cgi?acc=GSE138866). All data generated or analyzed

## Abstract

The long non-coding RNA (LncRNA) PAXIP1 antisense RNA 1 (PAXIP1-AS1) was found to promote proliferation, migration, EMT, and apoptosis of ovarian cancer (OC) cells in OC cell lines, but the relationship between PAXIP1-AS1 expression and clinical characteristics, prognosis, and immune infiltration of OC patients and its regulatory network are unclear. 379 OC tissues were collected from The Cancer Genome Atlas (TCGA) database. 427 OC tissues and 88 normal ovarian tissues were collected from GTEx combined TCGA database. 130 OC samples were collected from GSE138866. Kruskal-Wallis test, Wilcoxon sign-rank test, logistic regression, Kaplan-Meier method, Cox regression analysis, Gene set enrichment analysis (GSEA), and immuno-infiltration analysis were used to evaluate the relationship between clinical characteristics and PAXIP1-AS1 expression, prognostic factors, and determine the significant involvement of PAXIP1-AS1 in function. QRT-PCR was used to validate the expression of PAXIP1-AS1 in OC cell lines. Low PAXIP1-AS1 expression in OC was associated with age (P = 0.045), histological grade (P = 0.011), and lymphatic invasion (P = 0.004). Low PAXIP1-AS1 expression predicted a poorer overall survival (OS) (HR: 0.71; 95% CI: 0.55–0.92; P = 0.009), progression free interval (PFS) (HR: 1.776; 95% CI: 1.067–2.955; P = 0.001) and disease specific survival (DSS) (HR: 0.67; 95% CI: 0.51–0.89; P = 0.006). PAXIP1-AS1 expression (HR: 0.711; 95% CI: 0.542–0.934; P = 0.014) was independently correlated with PFS in OC patients. GSEA demonstrated that neutrophil degranulation, signaling by Interleukins, GPCR-ligand binding, G alpha I signaling events, VEGFAVEGFR-2 signaling pathway, naba secreted factors, Class A 1 Rhodopsin-Like Receptors, PI3K-Akt signaling pathway, and Focal Adhesion-PI3K-Akt-mTOR-signaling pathway were differentially enriched in PAXIP1-AS1 high expression phenotype. PAXIP1-AS1 was significantly downregulated in OC cell lines compared with IOSE29 cell line. The expression of PAXIP1-AS1 was associated with immune infiltration. low expression of PAXIP1-AS1 was correlated with poor OS (HR: 0.52; 95% CI: 0.34–0.80; P = 0.003) from GSE138866. There were some genomic variations between the PAXIP1-AS1 high and

during this study are included in this published article. The details of code and other original data were deposited in GitHub (https://github.com/BuzeChen15262020735/PAXIP1-AS1.git).

**Funding:** This work was supported by Xuzhou Key R&D Programme (ZYSB20210489) from Buze Chen. I had role in conceptualization, funding acquisition, writing – original draft, and writing – review & editing.

**Competing interests:** The authors have declared that no competing interests exist.

low expression groups. Low expression of PAXIP1-AS1 was significantly associated with poor survival and immune infiltration in OC. PAXIP1-AS1 could be a promising prognosis biomarker and response to immunotherapy for OC.

## Introduction

Ovarian cancer (OC) is one of the most dead malignancies in the female reproductive system [1]. Nearly 295,000 women worldwide have been diagnosed with OC and 185,000 have died from the disease [2]. Over 70% of OC patients are diagnosed at an advanced stage (FIGO stage III or IV) due to nonspecific symptoms in the early stages and the lack of effective screening methods [3]. The 5-year survival rate for stage III-IV patients is approximately 30% and for stage I patients is approximately 92% [4]. Despite considerable progress in some of these areas, the treatment of this tumor remains a major challenge in gynaecological oncology, with little change in long-term survival rates for HGSOC and several issues hindering progress in clinical outcomes.

Long noncoding RNAs (lncRNAs) comprise a class of RNA transcripts >200 nucleotides in length that act as key regulators of target gene expression in a variety of biological processes including chromatin modification, gene transcription, RNA splicing, RNA transport and translation [5]. Aberrant lncRNA expression may be critical for cancer development and progression, and lncRNA-mediated biology may be central in cancer development [6]. Unlike miRNAs and protein-encoding mRNAs, lncRNAs typically display restricted tissue-specific and cancer-specific expression patterns [7]. Furthermore, their expression is lower than that of protein-coding genes [8]. Given this lncRNA tissue specificity, they may be superior biomarkers to many current protein-coding biomarkers [9]. Therefore, screening for molecular markers associated with OC prognosis is important for the precise treatment of OC.

Changes in the expression of some lncRNAs have been reported in OC and in association with clinical characteristics [10, 11]. Silencing of the lncRNA PAXIP1-AS1, a key mediator of cell death, was found to contribute to cell survival [12]. Aberrant expression of the lncRNA PAXIP1-AS1 was evident in glioma cells and tissues and was significantly associated with survival outcomes in glioma patients [13]. H3K27ac-inducible lncRNA PAXIP1-AS1 promotes cell proliferation, migration, EMT, and apoptosis by targeting the miR-6744-5p/PCBP2 axis in OC [14]. However, the clinical significance of PAXIP1-AS1, its relevance to immune infiltration, and its regulatory network have not been investigated.

This study compared PAXIP1-AS1 expression differences between tumor tissues and normal samples based on The Cancer Genome Atlas (TCGA) database and OC RNA-seq data in GTEx, and assessed the correlation between PAXIP1-AS1 expression levels and clinical features of OC, and the prognostic value of PAXIP1-AS1 in OC. Genomic enrichment analysis (GSEA) was performed on the high and low PAXIP1-AS1 expression groups to reveal their possible functions. Correlation analysis between PAXIP1-AS1 expression and immune infiltration was performed to explore the potential mechanisms by which PAXIP1-AS1 regulates OC occurrence and development. This study provided a new direction for the individualized and precise treatment of OC.

## Materials and methods

### Clinical information

379 OC tissues with clinical information data were collected from TCGA database. Clinical information included clinical stage, primary treatment outcome, race, age, histological grade,

anatomical neoplasm subdivision, venous invasion, tumor status, lymphatic invasion, and tumor residual. The grouping was based on the median [15–17]. Details of the clinical information can be obtained from "OV_rnaseq_clinical_raw.xlsx" (https://github.com/BuzeChen15262020735/PAXIP1-AS1.git).

## Data processing

Data were RNAseq data in level 3 HTSeq-FPKM format from the TCGA (https://portal.gdc.cancer.gov/) OC project [18]. RNAseq data in fragments per million kilobases (FPKM) format were converted to transcripts per million reads (TPM) format and log2 for analysis [19, 20]. Supplementary data were prognostic data from a reference [21]. The data was filtered by retaining clinical information data.

## Differential expression of PAXIP1-AS1

427 OC tissues and 88 normal ovarian tissues were collected from GTEx combined TCGA database. UCSC XENA (https://xenabrowser.net/datapages/) TPM-formatted RNAseq data from TCGA and GTEx were processed uniformly by the Toil process [22]. Extracted OC data corresponding to TCGA and normal tissue data corresponding to GTEx. The data were not filtered. Data were processed as log2 (value+1).

The R package included ggplot2 [3.3.6], stats [4.2.1], and car [3.1–0]. The process was to select the appropriate statistical method for the statistics (stats package and car package) depending on the characteristics of the data format and to visualize the data using the ggplot2 package. The statistical method was the Welch t' test. The dependent variable was PAXIP1-AS1 [ENSG00000273344]. Details of the code can be obtained from "groupplot_gene.R" (https://github.com/BuzeChen15262020735/PAXIP1-AS1.git).

## The relationship between PAXIP1-AS1 and clinical characteristics

Logistic analysis. The R package was mainly the base package. The statistical method was the dichotomous logistic model. The dependent variable was PAXIP1-AS1. The type of independent variable was low high dichotomous. Details of the code can be obtained from " gene_logistic.R" (https://github.com/BuzeChen15262020735/PAXIP1-AS1.git).

Clinical correlation analysis. The R package was mainly ggplot2 [3.3.3]. Clinical variables included age, histological grade, and lymphatic invasion. Details of the codes can be obtained from "groupplot_age.R", groupplot_histological grade.R", and " groupplot_ lymphatic invasion.R" (https://github.com/BuzeChen15262020735/PAXIP1-AS1.git).

## The relationship between PAXIP1-AS1 and prognosis

Kaplan-Meier method analysis. The R package included the survminer package [0.4.9] and the survival package [3.2–10]. Subgroups were 0–50 and 50–100. The prognostic types were overall survival (OS), progression free survival (PFS), and disease specific survival (DSS). Details of the codes can be obtained from "kmplot_OS.R", "kmplot_PFS.R", and "kmplot_DSS.R" (https://github.com/BuzeChen15262020735/PAXIP1-AS1.git).

COX regression analysis. The prognostic type was PFS. Included variables were clinical features and PAXIP1-AS1. Details of the code can be obtained from "cox.R" (https://github.com/BuzeChen15262020735/PAXIP1-AS1.git).

We used the ggplot2 package to complete the forest plot.

Nomogram plot analysis. R package is the rms package [6.2–0] & survival package [3.2–10]. Prognosis type was PFS. Included variables were FIGO stage, primary therapy outcome, tumor

residual, and PAXIP1-AS1. Details of the code can be obtained from "nomogram_surv.R" (https://github.com/BuzeChen15262020735/PAXIP1-AS1.git).

### Gene set enrichment analysis (GSEA)

Single gene differential analysis. R package was DESeq2 [1.26.0] [23]. The target molecule was PAXIP1-AS1. Low expression group was 0–50%. The high expression group was 50–100%.

GSEA analysis. R package was mainly the clusterProfiler package [3.14.3] [24]. The method was Gene Set Enrichment Analysis (GSEA) analysis [25]. The seeds were Homo sapiens. Internal reference genes were collected as c2.cp.v7.2. symbols.gmt (curated). The generator database was a collection of MSigDB (database hyperlinks) (containing a detailed description of each generator). Details of GSEA original data can be obtained from "GSEA original data.xlsx" (https://github.com/BuzeChen15262020735/PAXIP1-AS1.git).

### Immune infiltration analysis by ssGSEA

The R package was the GSVA package [1.34.0] [26]. The immuno-infiltration algorithm was ssGSEA (built-in algorithm of the GSVA package). There were 24 immune cell markers which were from the reference [27]. Details of the code can be obtained from "immune_bbt.R" (https://github.com/BuzeChen15262020735/PAXIP1-AS1.git).

### QRT-PCR

The OC cell lines SKOV3 and A2780 and the human ovarian surface epithelial cell line IOSE29, which are maintained in our laboratory, were used in this study. Cells are cultured in RPMI-1640 medium supplemented with 10% fetal bovine serum (FBS) and 1% penicillin-streptomycin. All cells are stored in a humidified incubator with 5% $CO_2$ at 37˚C. The PAXIP1-AS1 levels in SKOV3, A2780, and IOSE29 cell lines were identified by qRT-PCR [28–30]. The primer sequences used are shown in Table 1. The thermal PCR profile was as follows: pre-denaturation at 94˚C for 3 minutes; denaturation at 94˚C 30 seconds, annealing at 50˚C for 40 seconds, extension at 72˚C for 1 minute, 30 cycles; extension at 72˚C for 10 minutes. ΔΔct is used to calculate qRT-PCR data. Using the same set of target gene-internal reference genes, which is the first Δct, Δct is subtracted from ΔΔct, and then we can calculate the exact value using the function = power (2, -ΔΔct).

### Correlation between PAXIP1-AS1 and prognosis in GSE138866

130 OC patients were collected from GSE138866. PAXIP1-AS1 expression data in patients were used to validate the prognostic value of PAXIP1-AS1. The R packages were the survminer package [0.4.9], and the survival package [3.2–10].

**Table 1. The sequence of primers in the present study.**

| Gene | | The sequence of primers (5'-3') |
|---|---|---|
| PAXIP1-AS1 | Forward | GAAGTTGGGAGAAGAAAT |
| | Reverse | AGTGTACCGCAGAGTAAT |
| U6 | Forward | CTCGCTTCGGCAGCACA |
| | Reverse | AACGCTTCACGAATTTGCGT |

## Genomic variation between PAXIP1-AS1 high and low expression groups

Patients with OC were grouped according to high or low PAXIP1-AS gene expression and their mutations were mapped oncoplot separately using the maftools package. Details of the code can be obtained from "Maftools.R" (https://github.com/BuzeChen15262020735/PAXIP1-AS1.git).

## Statistical analysis

All statistical analyses were performed using R (v.3.6.3) [31, 32]. Wilcoxon rank-sum test, chi-square test, and Fisher's exact test were used to analyze the relationship between clinical features and PAXIP1-AS1. P values less than 0.05 were considered statistically significant.

# Results

## Clinical characteristics

As shown in Table 2, the FIGO stage included 1 patient (0.3%), stage II in 23 (6.1%), stage III in 295 (78.5%), and stage IV in 57 (15.2%). The primary therapy outcome included 27 PD (8.8%), 22 SD (7.1%), 43 PR (14%), and 216 CR (70.1%). The race included 328 white patients, 12 Asian patients, and 25 Black or African American patients. The age included 208 patients ($< = 60$, 54.9%) and 171 patients ($>60$, 45.1%). The histologic grade included 1 G1 (0.3%), 45 G2 (12.2%), 322 G3 (87.3%), and 1 G4 (0.3%). The anatomic neoplasm subdivision 102 included unilateral (28.6%) and 255 bilateral (71.4%). The venous invasion included 64 yes (61%) and 41 no (39%). The lymphatic invasion included 101 yes (67.8%) and 48 no (32.2%). The tumor residual included 67 NRD (20%) and 268 RD (80%). The age range was 51 to 68 years, with a median of 59 years.

## PAXIP1-AS1 expression correlated with poor clinical characteristics of OC

PAXIP1-AS1 was low expressed in OC tissues (3.003±0.034 vs. 3.126±0.046, P = 0.032), based on 427 OC tissues and 88 normal ovarian tissues of GTEx combined TCGA database (Fig 1). The characteristics of OC patients were shown in Table 2, clinical and gene expression data were collected from TCGA database. According to the mean value of relative PAXIP1-AS1 expression, the patients with OC were divided into high (n = 190) and low (n = 189) expression groups. PAXIP1-AS1 expression was associated with age (P = 0.002), histological grade (P = 0.007), and lymphatic invasion (P = 0.007). As shown in Fig 2 and Table 3, PAXIP1-AS1 was significantly related to age (P = 0.045), histological grade (P = 0.011), and lymphatic invasion (P = 0.004).

## Relationship between PAXIP1-AS1 and survival of OC patients

As shown in Fig 3, expression of PAXIP1-AS1 was positively correlated with poor OS (HR: 0.71; 95% CI: 0.55–0.92; P = 0.009), PFS (HR: 1.776; 95% CI: 1.067–2.955; P = 0.001), and DSS (HR: 0.67; 95% CI: 0.51–0.89; P = 0.006) of OC patients. As shown in Table 4 and Fig 4, the results of univariate analysis showed low PAXIP1-AS1 expression levels were associated with worse PFS (HR: 1.776; 95% CI: 1.067–2.955; P = 0.001), primary therapy outcome (HR: 0.401; 95% CI: 0.304–0.528; P<0.001), and tumor residual (HR: 1.695; 95% CI: 1.219–2.358; P = 0.002). The result of multivariate analysis showed that PAXIP1-AS1 expression (HR: 0.711; 95% CI: 0.542–0.934; P = 0.014) and primary therapy outcome (HR: 0.496; 95% CI: 0.369–0.667; P<0.001) were independently correlated with PFS in multivariate analysis (Table 4). The results suggested that decreased expression of PAXIP1-AS1 level is associated with poor PFS. A nomogram was constructed to predict the 1-, 3-, and 5-year survival

**Table 2. Correlation between PAXIP1-AS1 expression and clinical characteristics in OC.**

| Characteristic | Overall | Low expression of PAXIP1-AS1 | High expression of PAXIP1-AS1 | P value |
|---|---|---|---|---|
| n | | 189 | 190 | |
| FIGO stage, n (%) | | | | 0.562 |
| Stage I | 1 (0.3%) | 0 (0%) | 1 (0.3%) | |
| Stage II | 23 (6.1%) | 9 (2.4%) | 14 (3.7%) | |
| Stage III | 295 (78.5%) | 148 (39.4%) | 147 (39.1%) | |
| Stage IV | 57 (15.2%) | 30 (8%) | 27 (7.2%) | |
| Primary therapy outcome, n (%) | | | | 0.409 |
| PD | 27 (8.8%) | 14 (4.5%) | 13 (4.2%) | |
| SD | 22 (7.1%) | 12 (3.9%) | 10 (3.2%) | |
| PR | 43 (14%) | 25 (8.1%) | 18 (5.8%) | |
| CR | 216 (70.1%) | 98 (31.8%) | 118 (38.3%) | |
| Race, n (%) | | | | 0.177 |
| Asian | 12 (3.3%) | 4 (1.1%) | 8 (2.2%) | |
| Black or African American | 25 (6.8%) | 9 (2.5%) | 16 (4.4%) | |
| White | 328 (89.9%) | 168 (46%) | 160 (43.8%) | |
| Age, n (%) | | | | 0.057 |
| < = 60 | 208 (54.9%) | 94 (24.8%) | 114 (30.1%) | |
| >60 | 171 (45.1%) | 95 (25.1%) | 76 (20.1%) | |
| Histologic grade, n (%) | | | | 0.007 |
| G1 | 1 (0.3%) | 1 (0.3%) | 0 (0%) | |
| G2 | 45 (12.2%) | 14 (3.8%) | 31 (8.4%) | |
| G3 | 322 (87.3%) | 170 (46.1%) | 152 (41.2%) | |
| G4 | 1 (0.3%) | 1 (0.3%) | 0 (0%) | |
| Anatomic neoplasm subdivision, n (%) | | | | 0.120 |
| Unilateral | 102 (28.6%) | 58 (16.2%) | 44 (12.3%) | |
| Bilateral | 255 (71.4%) | 120 (33.6%) | 135 (37.8%) | |
| Venous invasion, n (%) | | | | 0.107 |
| No | 41 (39%) | 15 (14.3%) | 26 (24.8%) | |
| Yes | 64 (61%) | 35 (33.3%) | 29 (27.6%) | |
| Lymphatic invasion, n (%) | | | | 0.007 |
| No | 48 (32.2%) | 15 (10.1%) | 33 (22.1%) | |
| Yes | 101 (67.8%) | 57 (38.3%) | 44 (29.5%) | |
| Tumor residual, n (%) | | | | 0.107 |
| NRD | 67 (20%) | 26 (7.8%) | 41 (12.2%) | |
| RD | 268 (80%) | 136 (40.6%) | 132 (39.4%) | |
| Age, median (IQR) | 59 (51, 68) | 61 (53, 71) | 57 (48.25, 66) | 0.002 |

probability of OC patients by combining the expression level of PAXIP1-AS1 with clinical variables, as shown in Fig 5.

## PAXIP1-AS1-related pathways based on GSEA

A dataset of 111 significant differences was enriched in PAXIP1-AS1 high expression phenotype. As shown in Table 5 and Fig 6, the top 9 low P-value datasets included neutrophil degranulation, signaling by interleukins, GPCR-ligand binding, G alpha I signaling events, VEGFA-VEGFR2 signaling pathway, secreted factors, Class A 1 Rhodopsin-Like Receptors, PI3K-Akt signaling pathway and focal adhesion-PI3K-Akt-mTOR-signaling pathway.

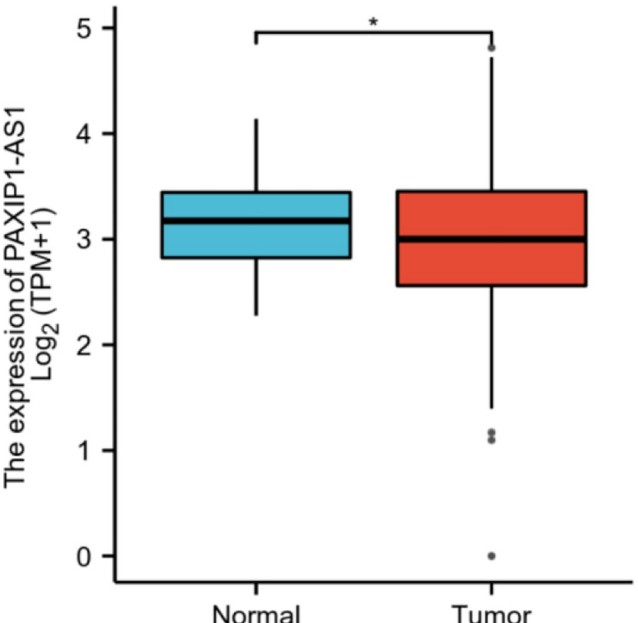

**Fig 1. Expression of PAXIP1-AS1 in OC and normal ovarian tissues.** *, P<0.05.

## Correlation of PAXIP1-AS1 expression with immune infiltration

For aDC, the mean level of PAXIP1-AS1 in the high expression group (0.398±0.147) was significantly lower than that in the low expression group (0.436±0.123) (P = 0.006) (Fig 7A). The correlation analysis (r = -0.110, P = 0.025) showed a negative correlation between PAXIP1-AS1 and aDC (Figs 8A and 9). For B cells, the mean level of PAXIP1-AS1 in the high expression group (0.174±0.07) was significantly lower than that in the low expression group (0.195±0.07) (P = 0.004) (Fig 7B). The correlation analysis (r = -0.190, P<0.001) showed a negative correlation between PAXIP1-AS1 and B cells (Figs 8B and 9). For CD8 T cells, the mean level of PAX-IP1-AS1 in the high expression group (0.606±0.021) was significantly lower than the mean level in the low expression group (0.611±0.021) (P = 0.013) (Fig 7C). The correlation analysis

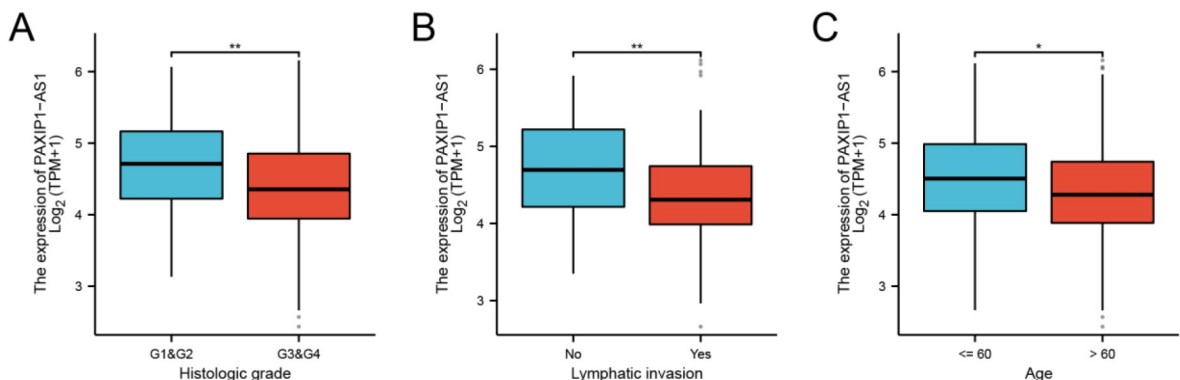

**Fig 2. Association with PAXIP1-AS1 expression and clinical stage.** (A) age. (B) histologic grade. (C) lymphatic invasion. *, P<0.05; **, P<0.01.

**Table 3. PAXIP1-AS1 expression associated with clinical characteristics (logistic regression).**

| Characteristics | Total (N) | Odds Ratio (OR) | P value |
|---|---|---|---|
| FIGO stage (Stage III & Stage IV vs. Stage I & Stage II) | 376 | 0.587 (0.241–1.353) | 0.220 |
| Primary therapy outcome (CR vs. PD&SD&PR) | 308 | 1.498 (0.918–2.455) | 0.107 |
| Race (White vs. Asian & Black or African American) | 365 | 0.516 (0.247–1.033) | 0.067 |
| Age (>60 vs. < = 60) | 379 | 0.660 (0.438–0.990) | 0.045 |
| Histological grade (G3 & G4 vs. G1 & G2) | 369 | 0.430 (0.218–0.815) | 0.011 |
| Anatomic neoplasm subdivision (Bilateral vs. Unilateral) | 357 | 1.483 (0.935–2.364) | 0.095 |
| Venous invasion (Yes vs. No) | 105 | 0.478 (0.211–1.058) | 0.072 |
| Lymphatic invasion (Yes vs. No) | 149 | 0.351 (0.166–0.715) | 0.005 |
| Tumor residual (RD vs. NRD) | 335 | 0.615 (0.353–1.057) | 0.082 |

(r = -0.130, P = 0.01) showed a negative correlation between PAXIP1-AS1 and CD8 T cells (Figs 8C and 9). For Cytotoxic cells, the mean level of PAXIP1-AS1 in the high expression group (0.358±0.108) was significantly lower than that in the low expression group (0.408 ±0.097) (P<0.001) (Fig 7D). The correlation analysis (r = -0.260, P<0.001) showed a negative correlation between PAXIP1-AS1 and Cytotoxic cells (Figs 8D and 9). For DC, the mean level of PAXIP1-AS1 in the high expression group (0.315±0.106) was significantly lower than that in the low expression group (0.34±0.108) (P = 0.027) (Fig 7E). The correlation analysis (r = -0.160, P = 0.002) showed a negative correlation between PAXIP1-AS1 and DC (Figs 8E and 9). For iDC, the mean level of PAXIP1-AS1 in the high expression group (0.401±0.064) was significantly lower than that in the low expression group (0.423±0.069) (P = 0.001) (Fig 7F). The correlation analysis (r = -0.230, P<0.001) showed a negative correlation between PAXI-P1-AS1 and iDC (Figs 8F and 9). For Macrophages, the mean level of PAXIP1-AS1 in the high expression group (0.518±0.065) was significantly lower than that of the low expression group (0.541±0.064) (P<0.001) (Fig 7G). The correlation analysis (r = -0.260, P<0.001) showed a negative correlation between PAXIP1-AS1 and Macrophages (Figs 8G and 9). For Mast cells, the mean level of PAXIP1-AS1 in the high expression group (0.122±0.062) was significantly lower than that of the low expression group (0.138±0.068) (P = 0.013) (Fig 7H). The correlation analysis (r = -0.160, P = 0.002) showed a negative correlation between PAXIP1-AS1 and Mast cells (Figs 8H and 9). For Neutrophils, the mean level of PAXIP1-AS1 in the high

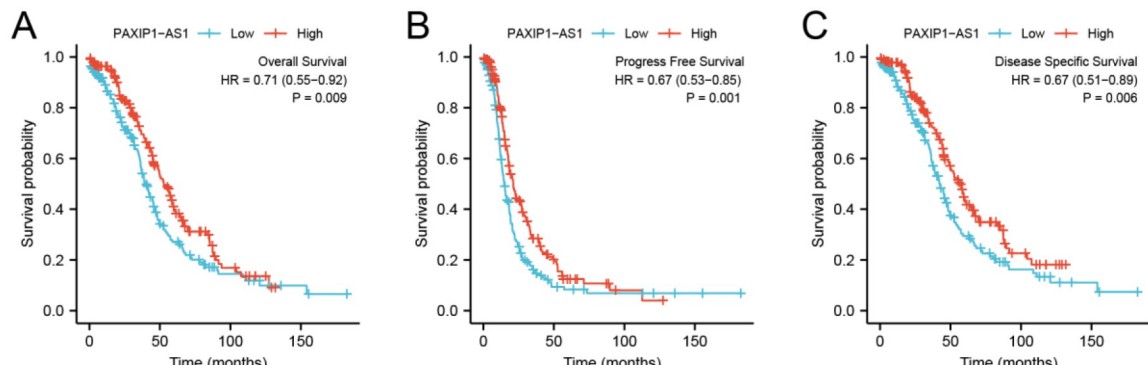

**Fig 3. Low expression of PAXIP1-AS1 was associated with poor OS, PFS, and DSS in OC patients.** (A) OS, over survival. (B) PFS, progress free survival. (C) DSS, disease specific survival.

**Table 4. Associations with PFS and clinical characteristics in TCGA OC patients (Cox regression).**

| Characteristics | Total (N) | Univariate analysis | | Multivariate analysis | |
|---|---|---|---|---|---|
| | | Hazard ratio (95% CI) | P value | Hazard ratio (95% CI) | P value |
| FIGO stage (Stage III & Stage IV vs. Stage I & Stage II) | 374 | 1.573 (0.918–2.694) | 0.099 | 1.516 (0.760–3.024) | 0.237 |
| Primary therapy outcome (CR vs. PD&SD&PR) | 307 | 0.401 (0.304–0.528) | <0.001 | 0.496 (0.369–0.667) | <0.001 |
| Race (White vs. Asian & Black or African American) | 364 | 0.843 (0.561–1.266) | 0.409 | | |
| Histologic grade (G3&G4 vs. G1&G2) | 367 | 1.188 (0.835–1.688) | 0.338 | | |
| Age (>60 vs. < = 60) | 377 | 1.076 (0.848–1.366) | 0.547 | | |
| Anatomic neoplasm subdivision (Bilateral vs. Unilateral) | 356 | 1.134 (0.865–1.488) | 0.363 | | |
| Venous invasion (No vs. Yes) | 105 | 0.890 (0.547–1.448) | 0.638 | | |
| Lymphatic invasion (Yes vs. No) | 148 | 1.115 (0.729–1.704) | 0.615 | | |
| Tumor residual (RD vs. NRD) | 334 | 1.695 (1.219–2.358) | 0.002 | 1.345 (0.928–1.949) | 0.117 |
| PAXIP1-AS1 (High vs. Low) | 377 | 0.668 (0.527–0.847) | <0.001 | 0.711 (0.542–0.934) | 0.014 |

expression group (0.233±0.075) was significantly lower than that of the low expression group (0.259±0.076) (P = 0.001) (Fig 7I). The correlation analysis (r = -0.200, P<0.001) showed a negative correlation between PAXIP1-AS1 and Neutrophils (Figs 8I and 9). For NK CD56dim cells, the mean level of PAXIP1-AS1 in the high expression group (0.12±0.09) was significantly lower than that of the low expression group (0.157±0.089) (P<0.001) (Fig 7J). The correlation analysis (r = -0.250, P<0.001) showed a negative correlation between PAXIP1-AS1 and NK CD56dim cells (Figs 8J and 9). For T cells, the mean level of PAXIP1-AS1 in the high expression group (0.281±0.124) was significantly lower than that of the low expression group (0.328 ±0.113) (P<0.001) (Fig 7K). The correlation analysis (r = -0.230, P<0.001) showed a negative correlation between PAXIP1-AS1 and T cells (Figs 8K and 9). For TFH, the mean level of PAXIP1-AS1 in the high expression group (0.311±0.038) was significantly lower than that of the low expression group (0.321±0.038) (P = 0.015) (Fig 7L). The correlation analysis (r = -0.140, P = 0.006) showed a negative correlation between PAXIP1-AS1 and TFH (Figs 8L and 9). For Tgd, the mean level of PAXIP1-AS1 in the high expression group (0.203±0.035) was significantly lower than that of the low expression group (0.219±0.039) (P<0.001) (Fig 7M). Correlation analysis (r = -0.260, P<0.001) showed a negative correlation between PAXIP1-AS1

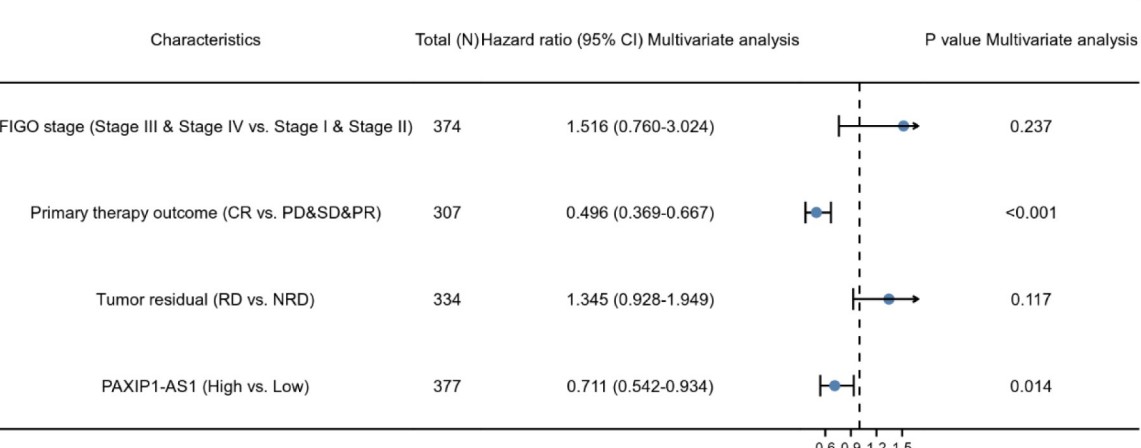

**Fig 4. Forest plot of the multivariate Cox regression analysis in OC.**

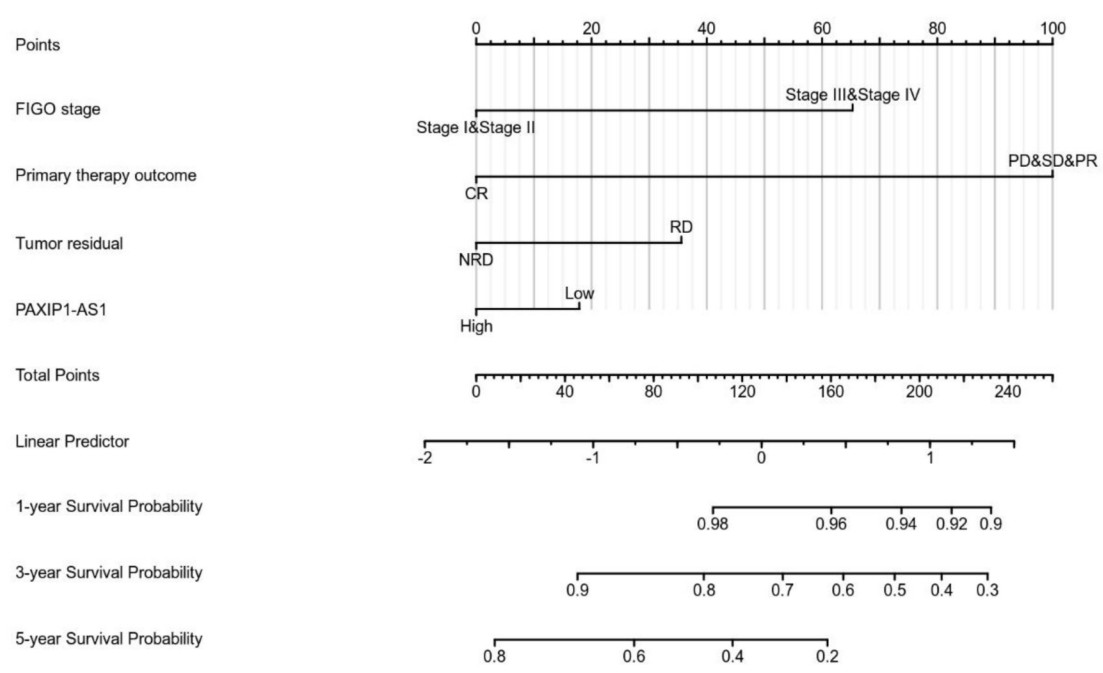

**Fig 5. Nomogram for predicting the probability of patients with 1-, 3- and 5-year OS.**

and Tgd (Figs 8M and 9). For Th1 cells, the mean level in the PAXIP1-AS1 high expression group (0.337±0.059) was significantly lower than the mean level in the low expression group (0.355±0.059) (P = 0.002) (Fig 7N). Correlation analysis (r = -0.180, P<0.001) showed a negative correlation between PAXIP1-AS1 and Th1 cells (Figs 8N and 9). For Th2 cells, the mean level of PAXIP1-AS1 in the high expression group (0.355±0.045) was significantly lower than that of the low expression group (0.369±0.04) (P = 0.002) (Fig 7O). Correlation analysis (r = -0.210, P<0.001) showed a negative correlation between PAXIP1-AS1 and Th2 cells (Figs 8O and 9). For TReg, the mean level in the PAXIP1-AS1 high expression group (0.297±0.138) was significantly lower than the mean level in the low expression group (0.341±0.135) (P = 0.002) (Fig 7P). Correlation analysis (r = -0.170, P = 0.001) showed a negative correlation between PAXIP1-AS1 and TReg (Figs 8P and 9).

**Table 5. Enrichment of gene sets in PAXIP1-AS1 high and low expression groups in OC (GSEA).**

| Gene set name | NES | P adjust | FDR |
|---|---|---|---|
| REACTOME_NEUTROPHIL_DEGRANULATION | -2.055 | 0.036 | 0.030 |
| REACTOME_SIGNALING_BY_INTERLEUKINS | -1.940 | 0.036 | 0.030 |
| REACTOME_GPCR_LIGAND_BINDING | -1.839 | 0.036 | 0.030 |
| REACTOME_G_ALPHA_I_SIGNALLING_EVENTS | -1.563 | 0.036 | 0.030 |
| WP_VEGFAVEGFR2_SIGNALING_PATHWAY | -1.570 | 0.036 | 0.030 |
| NABA_SECRETED_FACTORS | -1.962 | 0.036 | 0.030 |
| REACTOME_CLASS_A_1_RHODOPSIN_LIKE_RECEPTORS_ | -2.020 | 0.036 | 0.030 |
| WP_PI3KAKT_SIGNALING_PATHWAY | -1.840 | 0.036 | 0.030 |
| WP_FOCAL_ADHESIONPI3KAKTMTORSIGNALING_PATHWAY | -1.928 | 0.036 | 0.030 |

NES, normalized enrichment score; FDR, false discovery rate.

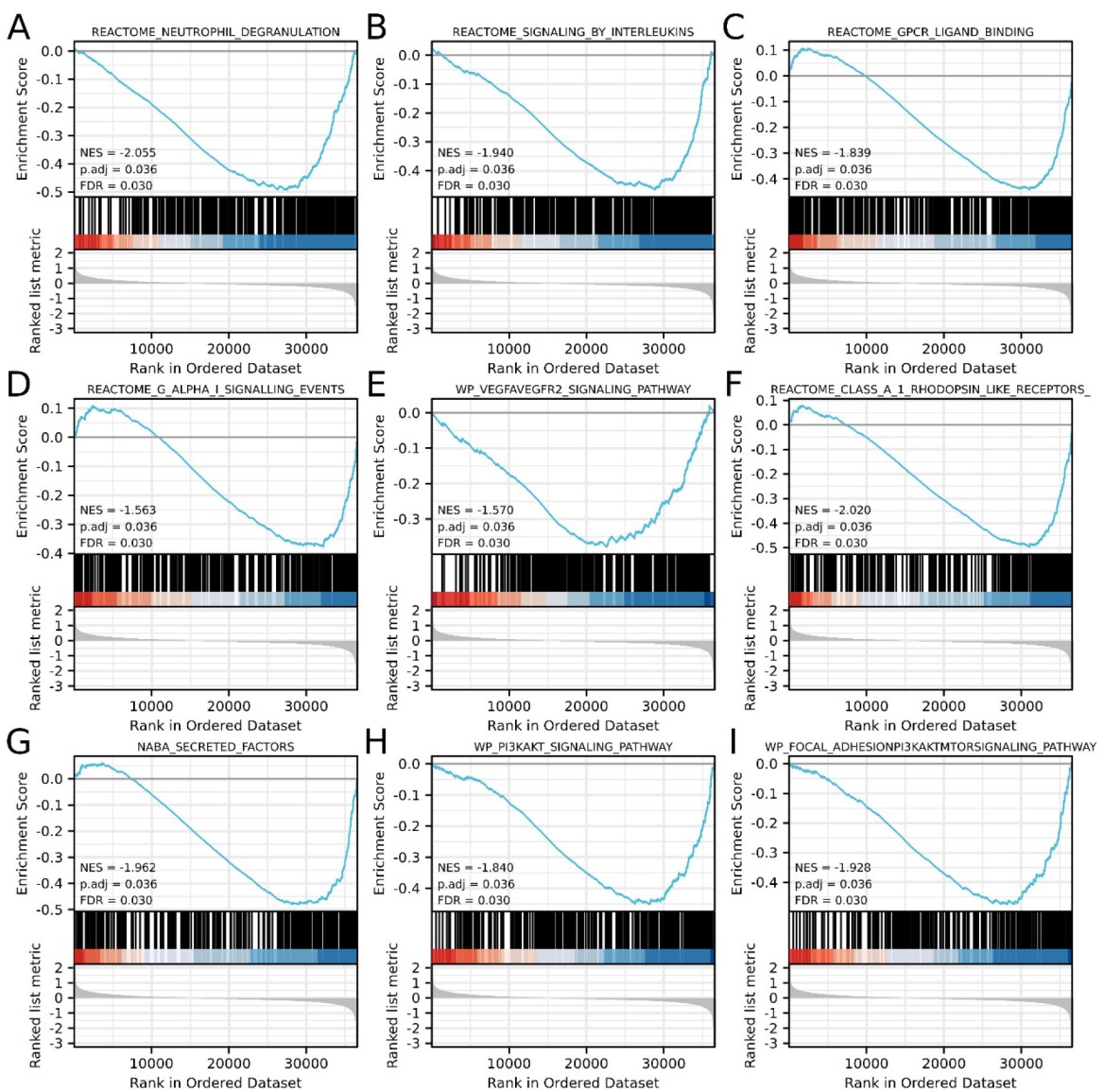

**Fig 6. GSEA analysis of PAXIP1-AS1 in OC.** (A) neutrophil degranulation, (B) signaling by Interleukins, (C) GPCR-ligand binding, (D) G alpha I signaling events, (E) VEGFA-VEGFR-2 signaling pathway, (F) Class A 1 Rhodopsin-Like Receptors, (G) secreted factors, (H) PI3K-Akt signaling pathway, and (I) Focal Adhesion-PI3K-Akt-mTOR-signaling pathway, were enriched in PAXIP1-AS1-related OC. NES, normalized ES; FDR, false discovery rate.

## Validation of PAXIP1-AS1 expression and prognostic value

The expression of PAXIP1-AS1 in SKOV3 was significantly lower than that in IOSE29 (0.209 ± 0.334 vs. 1.273 ± 0.291, P<0.05) (Fig 10). The expression of PAXIP1-AS1 in A2780 was significantly lower than that in IOSE29 (0.131 ± 0.316 vs. 1.273 ± 0.291, P<0.01) (Fig 10). These results suggested that PAXIP1-AS1 was significantly downregulated in OC cell lines compared with ovarian surface epithelial IOSE29 cell lines. As shown in Fig 11, low expression of PAXIP1-AS1 was correlated with poor OS (HR: 0.52; 95% CI: 0.34–0.80; P = 0.003).

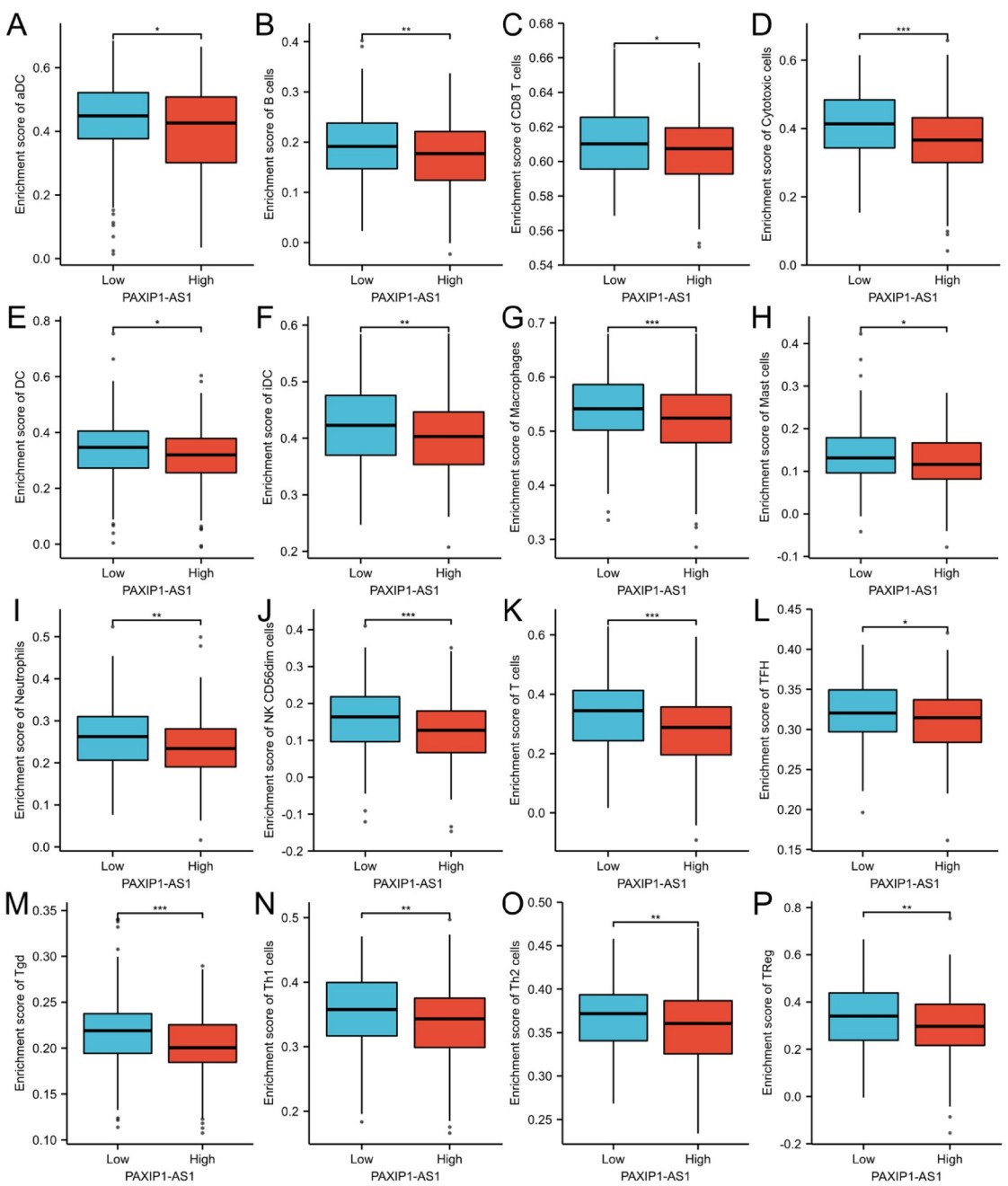

**Fig 7. Correlation between PAXIP1-AS1 expression and 24 immune cells in OC (grouped comparison plots).** (A) aDC, (B) B cells, (C) CD8 T cells, (D) cytotoxic cells, (E) DC, (F) iDC, (G) Macrophages, (H) Mast cells, (I) Neutrophils, (J) NK CD56dim cells, (K) T cells, (L) TFH, (M) Tgd, (N) Th1 cells, (O) Th2 cells, and (P) TReg.

## Comparison of genomic variation between PAXIP1-AS1 high and low expression groups

As shown in Fig 12, the top 10 shared genes with the highest mutation frequencies included TP53, TTN, CSMD3, USH2A, MUC16, MUC17, RYR2, HMCN1, DST, and FLG in the PAXIP1-AS1 high and low expression groups, As for CSMD3, the PAXIP1-AS1 high and low

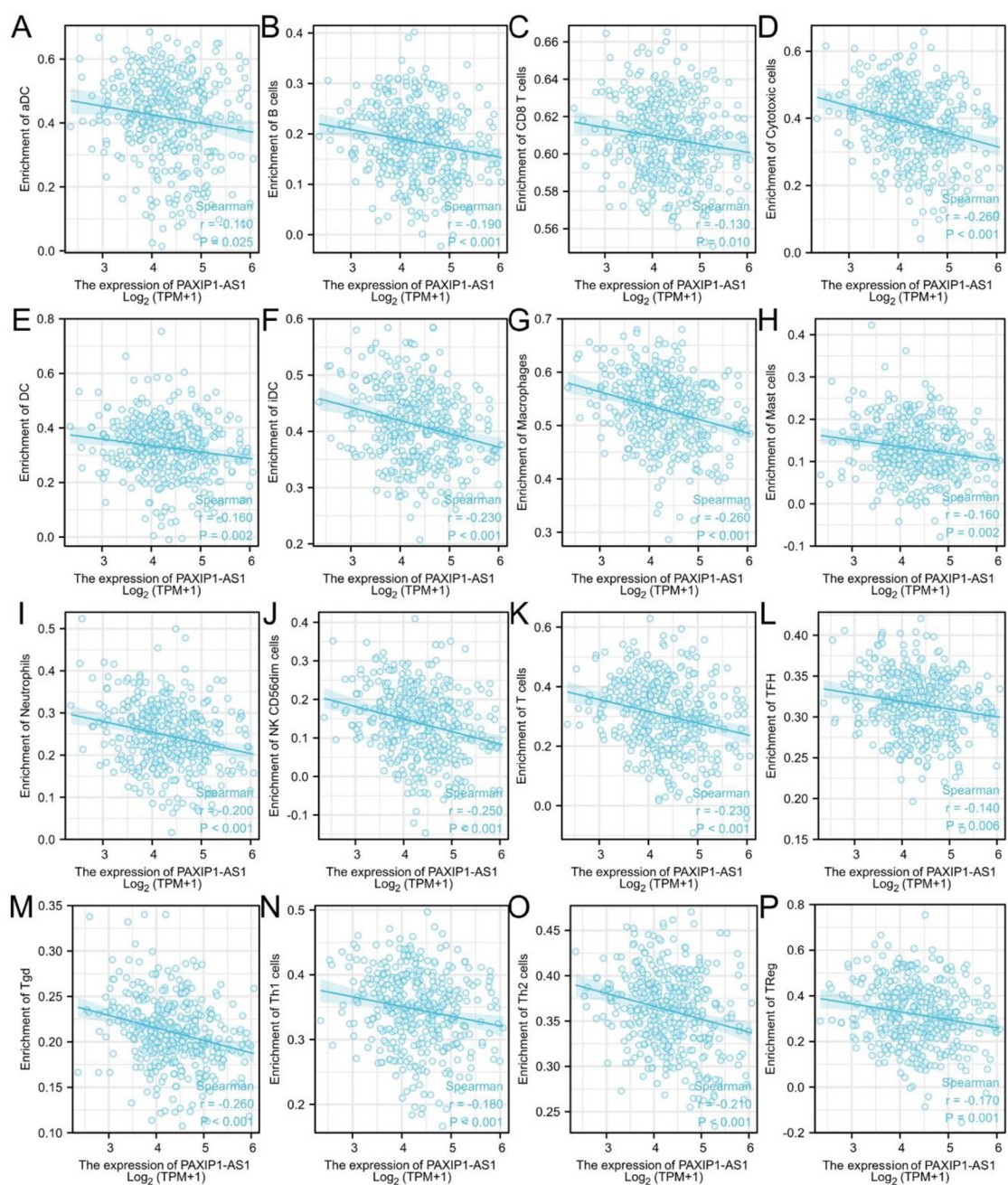

**Fig 8. Correlation between PAXIP1-AS1 expression and 24 immune cells in OC (scatter plots).** (A) aDC, (B) B cells, (C) CD8 T cells, (D) cytotoxic cells, (E) DC, (F) iDC, (G) Macrophages, (H) Mast cells, (I) Neutrophils, (J) NK CD56dim cells, (K) T cells, (L) TFH, (M) Tgd, (N) Th1 cells, (O) Th2 cells, and (P) TReg.

expression groups contained nonsense mutation and muti hit, respectively. As for USH2A, the PAXIP1-AS1 high and low expression groups contain nonsense mutation and splice site, and muti hit, respectively. As for MUC16, the PAXIP1-AS1 high expression group contains nonsense mutation compared to the PAXIP1-AS1 low expression group. As for MUC17, the PAXIP1-AS1 low expression group contains nonsense mutation compared to the PAXIP1-AS1

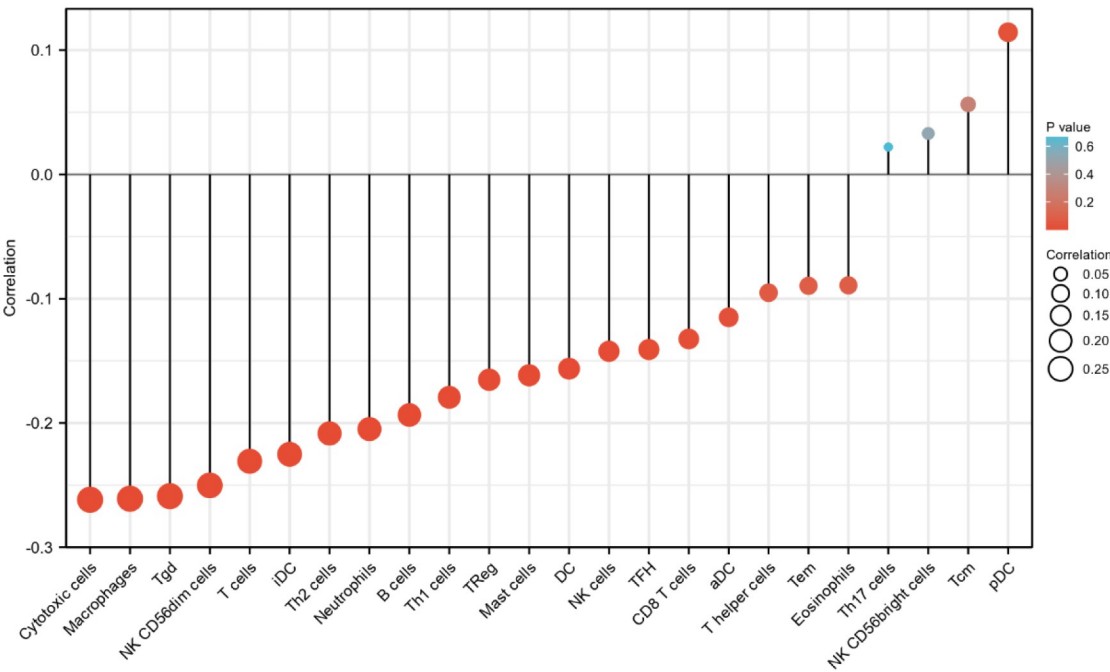

**Fig 9. Correlation between PAXIP1-AS1 expression level and 24 immune cells in OC (lollipop chart).** The size of the dots indicates the absolute value of Spearman r.

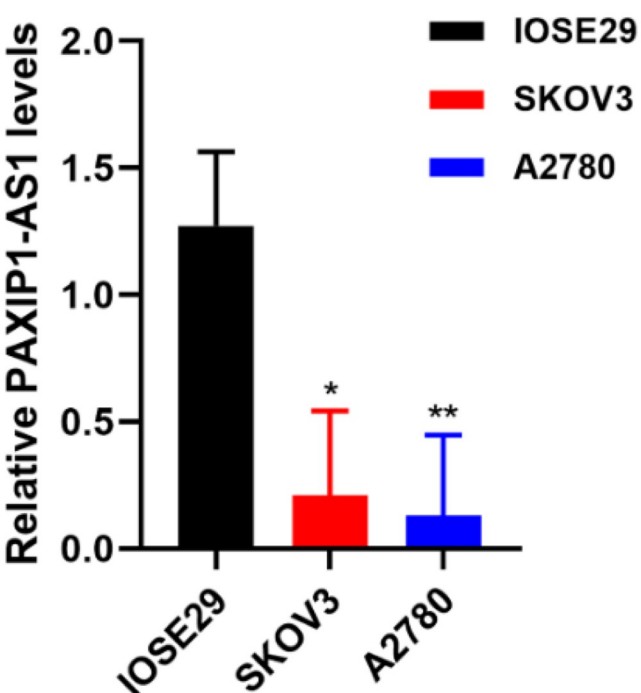

**Fig 10. Expression of PAXIP1-AS1 in ISOE29, SKOV3, andA2780 cell lines.** *, $P<0.05$; **, $P<0.01$.

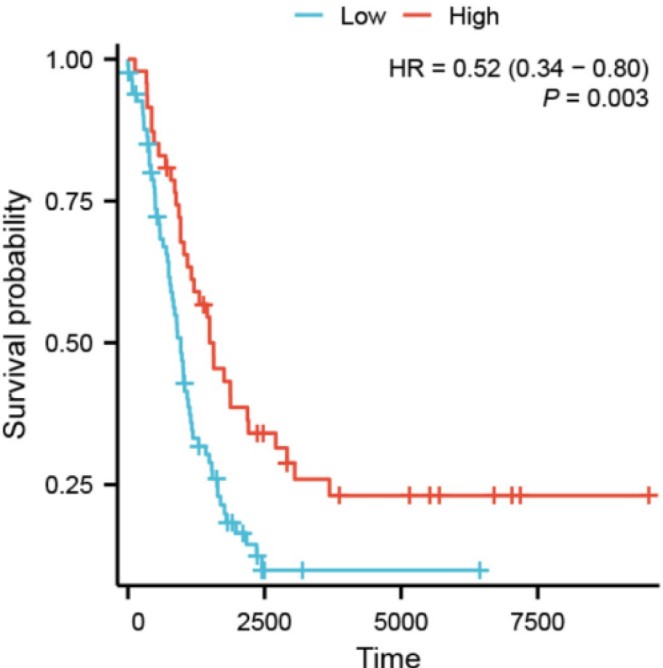

**Fig 11. Low expression of PAXIP1-AS1 was associated with poor OS in GSE138866.**

high expression group. As for RYR2, the PAXIP1-AS1 high expression group contains frame shift del compared to the PAXIP1-AS1 low expression group. As for HMCN1, the PAXIP1-AS1 high expression group contains nonsense mutation, frame shift del and muti hit compared to the PAXIP1-AS1 low expression group. As for DST, the PAXIP1-AS1 high expression

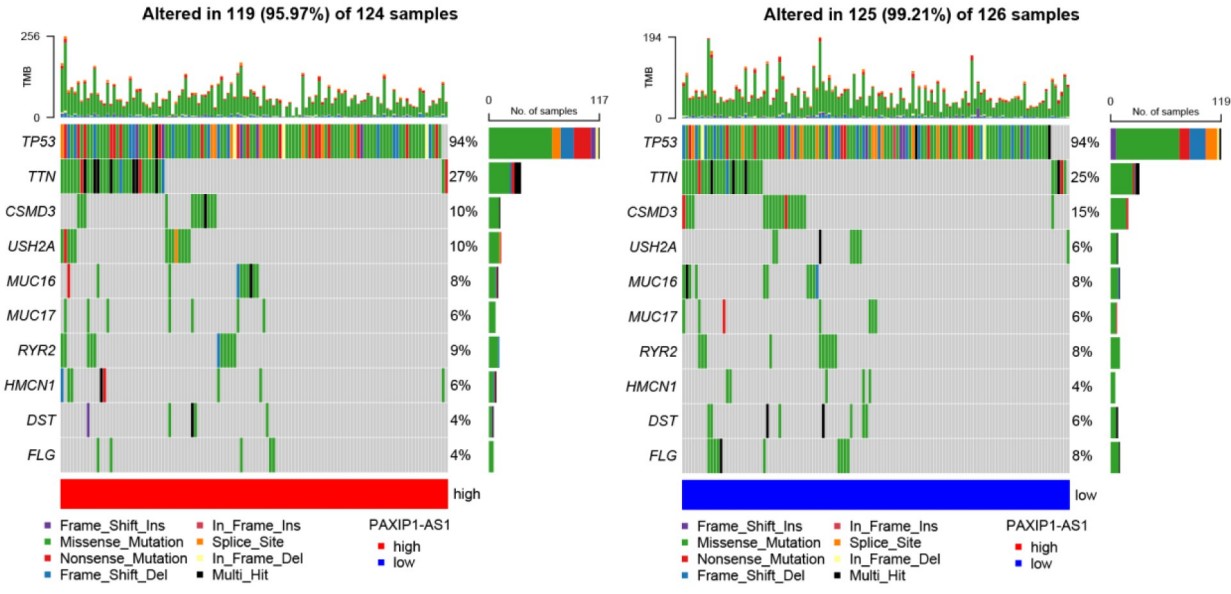

**Fig 12. Analysis of genomic variation between high and low expression groups of PAXIP1-AS1.**

group contains frame shift ins compared to the PAXIP1-AS1 low expression group. As for FLG, the PAXIP1-AS1 low expression group contains muti hit compared to the PAXIP1-AS1 high expression group. These results may help us to explore the possible mechanisms of PAXIP1-AS1-mediated ovarian carcinogenesis.

## Discussion

LncRNAs have been implicated in the molecular mechanisms of carcinogenesis [33]. As regulators of the flow of genetic information interacting with epigenetic, transcriptional, and post-transcriptional pathways, lncRNAs promote tumor formation, progression, and metastasis in many human malignancies [34]. Understanding the specific molecular events that underpin OC tumorigenesis can lead to early detection and improved outcomes. LOXL1-AS1 expression correlates with poor clinical outcome in EOC patients and can be used as an independent prognostic indicator as well as a new diagnostic biomarker [35]. LINC00472 may be a potential tumor suppressor in OS by interacting with miR-300 and FOXO1 [36]. High plasma levels of lncRNA ROR can be used as a potential biomarker for the diagnosis of OC [37]. Therefore, it is important to study lncRNAs as new prognosis OC biomarkers and therapeutic targets in the future.

High expression of PAXIP1-AS1 was observed in OC cell lines compared with HOSEpiC cell line, and exhibited an oncogenic role by facilitating cell proliferation, migration, EMT, and suppressing cell apoptosis [14]. In this study, the expression of PAXIP1-AS1 in OC were significantly lower than that in paired normal tissues, as well as in ovarian surface epithelial cells and OC cells. OC patients (age, > 60) showed significant low PAXIP1-AS1 expression. OC patients (G3 & G4) showed significant low PAXIP1-AS1 expression. OC patients (lymphatic invasion) showed significant low PAXIP1-AS1 expression. It indicated that patients with OC of high age, high differentiation, and lymphatic invasion have a poor prognosis. Expression of PAXIP1-AS1 was positively correlated with poor OS (P = 0.009), PFS (P = 0.001), and DSS (P = 0.006) of OC patients. PAXIP1-AS1 expression (HR: 0.711; 95% CI: 0.542–0.934; P = 0.014) was an independently correlated with PFS in OC patients. In conclusion, PAXIP1-AS1 is a good molecular marker of prognosis for patients with OC.

Overexpression of PAXIP1-AS1 advances glioma development by recruiting the transcription factor ETS1 to increase KIF14 expression [13]. PAXIP1-AS1 may modulate smooth muscle cell function by affecting multiple IPAH-specific transcriptional programs [38]. Based on GSEA, PAXIP1-AS1 was found to be involved in pathways including neutrophil degranulation, signaling by interleukins, GPCR-ligand binding, G alpha I signaling events, VEGFA-VEGFR2 signaling pathway, secreted factors, Class A 1 Rhodopsin-Like Receptors, PI3K-Akt signaling pathway, and focal adhesion-PI3K-Akt-mTOR-signaling pathway. The specific molecular mechanism by which PAXIP1-AS1 receives OC occurs needs further study.

Immune infiltration of OC is currently a hot topic and understanding of immune infiltration will facilitate the development of immunotherapy for OC. The results of this study showed a modest relationship between PAXIP1-AS1 expression and immune cells in OC. These correlations may suggest that PAXIP1-AS1 may inhibit the function of aDC, B cells, CD8 T cells, Cytotoxic cells, DC, iDC, Macrophages, Mast cells, Neutrophils, NK CD56dim cells, T cells, TFH, Tgd, Th1 cells, Th2 cells and Treg, which in turn exert a suppressive effect on OC through a potential mechanism. Differences in genomic variation between high and low PAXIP1-AS1 expression groups need to be further investigated for the mechanism of PAXIP1-AS1-mediated ovarian carcinogenesis.

Although there are some limitations, this is the further study to explore the relationship between PAXIP1-AS1 and OC. This study was mainly based on bioinformatic analysis and

could be further strengthened by experimental studies. The mechanism of PAXIP1-AS1-mediated ovarian carcinogenesis needs to be further investigated. Since there are some contradictory findings between this study and others, additional sample specimens, external validation cohorts, and specific biological experiments should be developed to improve the reliability. The innovation and reliability of this study needs to be improved.

## Conclusions

PAXIP1-AS1 showed low expression in OC tissues and cell lines. Low expression of PAXIP1-AS1 was related to poor OS, PFS, and DSS in OC patients. PAXIP1-AS1 might participate in the development of OC by pathways including neutrophil degranulation, signaling by interleukins, GPCR-ligand binding, G alpha I signaling events, VEGFA-VEGFR2 signaling pathway, secreted factors, Class A 1 Rhodopsin-Like Receptors, PI3K-Akt signaling pathway, and focal adhesion-PI3K-Akt-mTOR-signaling pathway. PAXIP1-AS1 expression was associated with immune infiltrating cells. There were some genomic variations between the PAXIP1-AS1 high and low expression groups. This study partly revealed that PAXIP1-AS1 could be a promising prognosis biomarker and response to immunotherapy for OC.

## Acknowledgments

The authors thank TCGA and GEO databases for providing the data. We state that a preprint of this paper has previously been published [39].

## Author Contributions

**Conceptualization:** Buze Chen, Hui Liu.

**Data curation:** Xiaoyuan Lu, Qingmei Zhou, Qing Chen, Siyan Zhu.

**Funding acquisition:** Buze Chen.

**Methodology:** Buze Chen, Hui Liu.

**Validation:** Xiaoyuan Lu, Qingmei Zhou, Qing Chen, Siyan Zhu.

**Writing – original draft:** Buze Chen, Xiaoyuan Lu, Guilin Li, Hui Liu.

**Writing – review & editing:** Buze Chen, Xiaoyuan Lu, Qingmei Zhou, Qing Chen, Siyan Zhu, Guilin Li, Hui Liu.

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
