## [Decision Letter · Decision Letter 0]

7 Feb 2023

PONE-D-23-01712PAXIP1-AS1 is associated with immune infiltration and predicts poor prognosis in ovarian cancerPLOS ONE

Dear Dr. Chen,

Thank you for submitting your manuscript to PLOS ONE. After careful consideration, we feel that it has merit but does not fully meet PLOS ONE’s publication criteria as it currently stands. Therefore, we invite you to submit a revised version of the manuscript that addresses the points raised during the review process.

We look forward to receiving your revised manuscript.

Kind regards,

Gurudeeban Selvaraj

Academic Editor

PLOS ONE

Journal Requirements:

"This work was supported by Xuzhou Key R&D Programme (ZYSB20210489) from Buze Chen."

"The authors declare that they have no competing interests. "

Additional Editor Comments:

In this paper, the authors comprehensively analyzed the effect of PAXIP1-AS1 expression on the prognosis of patients using OC gene expression and clinical information from the TCGA database, and verified the results by cell lines. In general, the author 's article box is clear, but the description and writing are confusing. The specific evaluation is as follows:

1. there are multiple OC datasets in the database, and it is recommended that the authors validate the results in other datasets.

2. The author 's method section is not clearly described, for example, it is simpler to describe in the data collection and processing stages.

3. The format and typesetting of the whole article are very poor, and the size ratio of the picture is inconsistent.

4. In Figure 3, the tail of the survival curve clearly crossed, how the authors explained it.

5. In Figure 9, PAXIP1-AS1 did not correlate highly with a variety of immune cells, all below 0.3.

6. It is well-known that the prognosis of tumors is more related to genomic variants, and whether the authors analyzed differences in genomic variants between patients in the high and low PAXIP1-AS1 expression groups.

Reviewers' comments:

Reviewer's Responses to Questions

**Comments to the Author**

1. Is the manuscript technically sound, and do the data support the conclusions?

Reviewer #1: Yes

2. Has the statistical analysis been performed appropriately and rigorously? 

Reviewer #1: Yes

3. Have the authors made all data underlying the findings in their manuscript fully available?

Reviewer #1: Yes

4. Is the manuscript presented in an intelligible fashion and written in standard English?

Reviewer #1: No

5. Review Comments to the Author

Reviewer #1: In this paper, the authors comprehensively analyzed the effect of PAXIP1-AS1 expression on the prognosis of patients using OC gene expression and clinical information from the TCGA database, and verified the results by cell lines. In general, the author 's article box is clear, but the description and writing are confusing. The specific evaluation is as follows:

1. there are multiple OC datasets in the database, and it is recommended that the authors validate the results in other datasets.

2. The author 's method section is not clearly described, for example, it is simpler to describe in the data collection and processing stages.

3. The format and typesetting of the whole article are very poor, and the size ratio of the picture is inconsistent.

4. In Figure 3, the tail of the survival curve clearly crossed, how the authors explained it.

5. In Figure 9, PAXIP1-AS1 did not correlate highly with a variety of immune cells, all below 0.3.

6. It is well-known that the prognosis of tumors is more related to genomic variants, and whether the authors analyzed differences in genomic variants between patients in the high and low PAXIP1-AS1 expression groups.

6. PLOS authors have the option to publish the peer review history of their article (what does this mean?). If published, this will include your full peer review and any attached files.

Reviewer #1: No

---

## [Author Response · Author response to Decision Letter 0]

7 Mar 2023

Dear Editor:

We have carefully studied the valuable comments from reviewers and the editor, and tried our best to revise the manuscript titled “PAXIP1-AS1 is associated with immune infiltration and predicts poor prognosis in ovarian cancer” (PONE-D-23-01712). The amendment has been marked with red. We have made detailed changes throughout the text. The point to point responds to the reviewer’s comments are listed in the revised text.

Reviewer #1: 

In this paper, the authors comprehensively analyzed the effect of PAXIP1-AS1 expression on the prognosis of patients using OC gene expression and clinical information from the TCGA database, and verified the results by cell lines. In general, the author 's article box is clear, but the description and writing are confusing. The specific evaluation is as follows:

1. there are multiple OC datasets in the database, and it is recommended that the authors validate the results in other datasets.

Response: Thank you for your comments. GSE138866 was used to validate the prognostic value of PAXIP1-AS1 according to your suggestion.

2. The author 's method section is not clearly described, for example, it is simpler to describe in the data collection and processing stages.

Response: Thank you for your comments. Method section was improved according to your suggestion.

3. The format and typesetting of the whole article are very poor, and the size ratio of the picture is inconsistent.

Response: Thank you for your comments. The formatting and layout of the full text has been improved in accordance with the requirements of the journal.

4. In Figure 3, the tail of the survival curve clearly crossed, how the authors explained it.

Response: Thank you for your comments. The Logrank test does not require the proportional risk assumption to be met and can be used on an ad hoc basis when the proportional risk assumption is not met. The Logrank test was used in Figure 3 in this study. GSE138866 was used to validate the prognostic value of PAXIP1-AS1 (Figure 11) according to your suggestion.

5. In Figure 9, PAXIP1-AS1 did not correlate highly with a variety of immune cells, all below 0.3.

Response: Thank you for your comments. As you say, the absolute values of the correlation coefficients were all less than 0.3. This suggested a weak and positive or negative correlation between PAXIP1-AS1 and the infiltration of immune cells by both variants. However, a statistically significant association of PAXIP1-AS1 with the infiltration of immune cells could be determined on the basis of p-values. The description of the text was improved according to your suggestion.

6. It is well-known that the prognosis of tumors is more related to genomic variants, and whether the authors analyzed differences in genomic variants between patients in the high and low PAXIP1-AS1 expression groups.

Response: Thank you for your comments. The genomic variants between patients in the high and low PAXIP1-AS1 expression groups were supplemented according to your suggestion.

Thank you for considering our manuscript.

Yours sincerely,

Buze Chen

---

## [Decision Letter · Decision Letter 1]

11 Apr 2023

PONE-D-23-01712R1PAXIP1-AS1 is associated with immune infiltration and predicts poor prognosis in ovarian cancerPLOS ONE

Dear Dr. Chen,

Thank you for submitting your manuscript to PLOS ONE. After careful consideration, we feel that it has merit but does not fully meet PLOS ONE’s publication criteria as it currently stands. Therefore, we invite you to submit a revised version of the manuscript that addresses the points raised during the review process.

We look forward to receiving your revised manuscript.

Kind regards,

Gurudeeban Selvaraj

Academic Editor

PLOS ONE

Additional Editor Comments:

It seems the authors of the manuscript ID PONE-D-23-01712R1 responded to all the questions raised by the reviewers. However, I found some weaknesses in the methodology section as follows,

Clinical information - No clinical info here, just data retrieval from TCGA; Data processing - No proper description, just data type conversion, how???; Differential expression of PAXIP1-AS1; The relationship between PAXIP1-AS1 and clinical characteristics; The relationship between PAXIP1-AS1 and prognosis; Gene set enrichment analysis (GSEA); Immune infiltration analysis by ssGSEA; Genomic variation between PAXIP1-AS1 high and low expression groups; Correlation between PAXIP1-AS1 and prognosis in GSE138866;----------- All the parameters, inputs, and output details need to be crystal clear.

The specific steps were performed according to the reference [33, 34]----describe briefly.

R software and statistical analysis - No proper description, just the name of the packages, meaningless. I suggest the author need to enclose all the scripts as an appendix or deposit them in GitHub if no copyright issues.

Reviewers' comments:

Reviewer's Responses to Questions

**Comments to the Author**

1. If the authors have adequately addressed your comments raised in a previous round of review and you feel that this manuscript is now acceptable for publication, you may indicate that here to bypass the “Comments to the Author” section, enter your conflict of interest statement in the “Confidential to Editor” section, and submit your "Accept" recommendation.

Reviewer #1: All comments have been addressed

2. Is the manuscript technically sound, and do the data support the conclusions?

Reviewer #1: Yes

3. Has the statistical analysis been performed appropriately and rigorously? 

Reviewer #1: Yes

4. Have the authors made all data underlying the findings in their manuscript fully available?

Reviewer #1: Yes

5. Is the manuscript presented in an intelligible fashion and written in standard English?

Reviewer #1: Yes

6. Review Comments to the Author

Reviewer #1: The author responded to all the questions. I have no more questions at the moment. Wish the author good luck.

7. PLOS authors have the option to publish the peer review history of their article (what does this mean?). If published, this will include your full peer review and any attached files.

Reviewer #1: No

---

## [Author Response · Author response to Decision Letter 1]

4 May 2023

Dear Editor:

We have carefully studied the valuable comments from the editor, and tried our best to revise the manuscript titled “PAXIP1-AS1 is associated with immune infiltration and predicts poor prognosis in ovarian cancer” (PONE-D-23-01712R1). The amendment has been marked with red. We have made detailed changes throughout the text. The point to point responds to the editor’s comments are listed in the revised text.

Additional Editor Comments:

Clinical information - No clinical info here, just data retrieval from TCGA; Data processing - No proper description, just data type conversion, how???; Differential expression of PAXIP1-AS1; The relationship between PAXIP1-AS1 and clinical characteristics; The relationship between PAXIP1-AS1 and prognosis; Gene set enrichment analysis (GSEA); Immune infiltration analysis by ssGSEA; Genomic variation between PAXIP1-AS1 high and low expression groups; Correlation between PAXIP1-AS1 and prognosis in GSE138866;----------- All the parameters, inputs, and output details need to be crystal clear.

Response: Thank you for your comments. The Materials and Methods section was improved according to your suggestion.

The specific steps were performed according to the reference [33, 34]----describe briefly.

Response: Thank you for your comments. The Materials and Methods section was improved according to your suggestion.

R software and statistical analysis - No proper description, just the name of the packages, meaningless. I suggest the author need to enclose all the scripts as an appendix or deposit them in GitHub if no copyright issues.

Response: Thank you for your comments. The Materials and Methods section was improved and the details of code and other original data were deposited in GitHub (https://github.com/BuzeChen15262020735/PAXIP1-AS1.git) according to your suggestion.

Thank you for considering our manuscript.

Yours sincerely,

Buze Chen

---

## [Editor Report · Decision Letter 2]

1 Aug 2023

PAXIP1-AS1 is associated with immune infiltration and predicts poor prognosis in ovarian cancer

PONE-D-23-01712R2

Dear Dr. Chen,

We’re pleased to inform you that your manuscript has been judged scientifically suitable for publication and will be formally accepted for publication once it meets all outstanding technical requirements.

Kind regards,

Gurudeeban Selvaraj

Academic Editor

PLOS ONE
---

## [Editor Report · Acceptance letter]

3 Aug 2023

PONE-D-23-01712R2 

PAXIP1-AS1 is associated with immune infiltration and predicts poor prognosis in ovarian cancer 

Dear Dr. Chen:

I'm pleased to inform you that your manuscript has been deemed suitable for publication in PLOS ONE. Congratulations! Your manuscript is now with our production department. 

Kind regards, 

on behalf of

Dr. Gurudeeban Selvaraj 

Academic Editor

PLOS ONE